# Visual Aesthetic Quality Assessment of Urban Forests: A Conceptual Framework

Riyadh Mundher [1,*], Shamsul Abu Bakar [1], Marwah Al-Helli [2], Hangyu Gao [1], Ammar Al-Sharaa [3], Mohd Johari Mohd Yusof [1], Suhardi Maulan [1] and Azlizam Aziz [4]

1   Department of Landscape Architecture, Faculty of Design and Architecture, Universiti Putra Malaysia, Serdang 43400, Malaysia
2   The Doctoral School of Architecture, "Ion Mincu" University of Architecture and Urban Planning, 010014 București, Romania
3   Department of Architecture, Faculty of Built Environment, University of Malaya, Kuala Lumpur 50603, Malaysia
4   Department of Recreation and Ecotourism, Faculty of Forestry and Environment, Universiti Putra Malaysia, Serdang 43400, Malaysia
*   Correspondence: arch.riyad@gmail.com

**Abstract:** Visual aesthetic quality is the visual pleasure level that attracts people and makes them prefer certain areas. Visual aesthetic quality is valued and considered for urban forests but remains challenging. This could be due to a lack of understanding of visual aesthetic quality assessment variables based on visual aesthetic theories. This study supports an integrated conceptual framework based on the result of a systematic literature review study to describe and measure aesthetics that incorporates objective and subjective factors through urban forest visual character and urban forest visual quality. The results include defining and understanding a description of visual aesthetic factors and variables as well as a thorough explanation of visual aesthetic theories to comprehend how to assess the visual aesthetic quality of urban forests. This study agrees with and supports the visual aesthetic theoretical framework, and we believe that due to our shared evolutionary history, humans have a standard set of urban forest visual aesthetic features with preferences that change according to cultural and personal variances. Furthermore, this research provides a foundation of visual aesthetic variables of urban forests that will assist urban forest researchers, urban forest managers, and decision-makers in managing and protecting the visual aesthetics of urban forests.

**Keywords:** aesthetic; aesthetic philosophy; aesthetic theory; forestry; urban forest; urban-forest visual character; urban-forest visual quality; urban green

## 1. Introduction

Environmental issues are becoming increasingly severe as urban green spaces decline; the value of landscaping, especially forests within urban settings, must therefore be recognised. Urban forests are of important value, and continue to gain importance due to their direct impact on the lives of urban residents. The value of urban forests lies not only in economic or environmental factors, but also in visual aesthetics [1]. The visual aesthetics of forests are of great value and are considered in both environmental and social research and management. Many public and researchers around the world value the visual aesthetics of forests for their significant impact on areas as diverse as well-being, health, and the economy [2]. The visual aesthetics of urban forests plays a significant role in improving the physical health of urban residents, producing benefits such as reduced stress, improved physical well-being for the elderly, enhanced disease recovery, walking motivation, improved attention capacity, a sense of good health and satisfaction, physical activity, and behavioral changes [2–4]. In addition to providing locals with a natural experience in the city, the visual aesthetics of an urban forest can play a crucial role in ecotourism, as a

tourist destination [2,5]. Aesthetic aspects influence tourists' experience and happiness, enhancing their loyalty and desire to return. When planning a trip, individuals primarily seek out venues with aesthetic attributes that enhance their pleasure [2,6]. Urban forest aesthetics can indirectly increase tourism and promote economic growth by contributing to an aesthetically pleasing green city [2]. In addition, urban forest aesthetic benefits are manifested in rises in property value, encouraging homeowners near forests to advocate forest aesthetic conservation due to the impact on real estate values [2,7]. Therefore, improving the aesthetic value of the urban forest is advantageous for physical health and well-being, attracting tourists, and boosting the local economy.

Many studies on visual aesthetics have been undertaken since the 1960s, including qualitative studies to discover aesthetic theories and philosophy and quantitative studies to measure visual aesthetic quality and public preferences and perception [8–11]. In addition to the most recent evaluation methodologies, Daniel, and Lothian [12,13] have offered useful broad research for the concept and philosophy of landscape aesthetics. They underlined the importance of the aesthetic quality of landscapes as one of the fundamental characteristics of human interaction with nature to convey a sense of pleasure to people. Thus, landscape aesthetics as a model for natural forests within urban settings and is defined as "a sensation of pleasure attributable to visually perceivable characteristics of spatially arranged landscape patterns" [14]. Previous studies defined the forest aesthetic component using terms such as "landscape aesthetic quality", "landscape visual quality", and "forest beauty" [12,13,15,16]. In this study, the definitions of terms working are provided in Box 1.

Aesthetic assessments are generally directly connected to perceptual experiences of certain environmental views and can be frequently visually assessed. Other perception inputs can also facilitate and govern aesthetic preferences such as emotional or cognitive perception. Notably, aesthetic preferences are primarily determined by emotional processes, which are influenced by changing physiological and psychological preferences and experiences [17]. Some researchers argue that pleasure derived from the ecological functions of landscapes can increase aesthetic preferences [14,18]. At the same time, others argue that political, cultural, philosophical and ethical ideals influence how people are drawn to and prefer forest aesthetics [19,20]. While aesthetic quality is a crucial indicator and measure of urban green areas such as urban forests, the question of how to assess aesthetics persists [1,21]. In order to arrive at an assessment of visual aesthetics, a definition and an understanding of the variables that go into determining the visual aesthetic quality of urban forests must be established. Furthermore, a broad understanding of the mechanisms that govern the effects of the variables that go into determining the visual aesthetic quality of urban forests must be considered, only then a comprehensive assessment of visual aesthetics can be achieved.

**Box 1.** Working definitions of terms used in this research.

---

- **Visual Aesthetic** is a sensation of pleasure attributable to the visually perceptible characteristics of spatial elements in a scene.
- **Urban Forest** is all the natural forests, planted forests, permanent reserves, and all associated vegetation growing near or within highly populated urban areas.
- **Urban-Forest Visual Character (UFVC)** is a distinct, recognisable, and consistent pattern of scene elements that makes one scene different from another, rather than better or worse.
- **Urban-Forest Visual Quality (UFVQ)** is the relative aesthetic excellence of the forest and is typically measured in terms of viewer appreciation of the scenery.

---

## 2. Visual Aesthetic Assessment Framework of Urban Forests Based on Aesthetic Philosophy

Philosophy is the pursuit of the ultimate reality to identify and describe. Philosophy investigates concepts, and it does so generally independently of experience. Aesthetics has been a philosophical topic at least since the time of Socrates (469–399 B.C.). The focus of philosophical inquiry was beauty until 1750, when the term aesthetics was coined by the German philosopher Alexander Baumgarten. However, after the term aesthetics was coined,

philosophy expanded its focus to include this broader term. Philosophers distinguish between the aesthetic object, the aesthetic recipient, and the aesthetic experience. The aesthetic object is that which evokes an experience in the recipient. Landscapes, including forests, are among the numerous aesthetic objects that philosophy has considered; with regard to human interaction with aesthetic objects, philosophers have sought to identify the common principles that guide and determine the aesthetic experience.

We have adopted a broad definition of the visual aesthetic experience of urban forests as a pleasurable emotion that is mainly perceived by visually perceptible characteristics of the spatial elements in a scene. However, due to philosophical differences, some argue that the pleasure of beauty is essential and present in urban forest elements. This philosophy is known as the objectivist paradigm, in which "visual aesthetic quality is an inherent feature of nature" and is assessed by applying criteria to the scene. Socrates, Plato, and Aristotle defined beauty in this paradigm as "giving pleasure when seen", saying that beauty exists within an individual, is not subject to prejudice by observers, and does not depend on the individual's experience or cultural background [6,13,22]. In other words, beauty is absolute, not relative. In addition, the aesthetics of Augustine (354–430) were based on the concepts of unity, number, equality, proportion, and order. He believed that the order and proportion of an object determined its unity, which means that beauty was not relative. According to Thomas Aquinas (1224–1274), beauty is a subset of goodness. The components of beauty are "integrity or perfection", "proper proportion or harmony", and "brightness or clarity". Bonaventure (1217–1274) viewed nature as a "mirror of God" that reflected God's perfection in varying degrees. Alberti (1404–1472) believed that beauty derives from order and arrangement, such that nothing can be altered except for the worse [13]. In this regard, Chinese philosophers Lao Tzu and Chuang Tzu explain aesthetics by achieving an invisible semblance aesthetic. In order to achieve the aesthetic, one must purge oneself of inner desire and external disturbance, maintain simplicity, abandon knowledge and wisdom, and forget everything in order to contact the natural law with nature, thereby achieving the essence of beauty apart from all external factors or belonging to the recipient [23]. In urban forests, some of the supporters of the idea that beauty is intrinsic confirmed that the reported cross-cultural differences in views of urban forests had been inflated more than might be expected because it was mixed with buildings. Likewise, other research has confirmed a high degree of similarity between aesthetic preferences in locations with people of different cultural backgrounds, despite disparities in demographic characteristics such as culture and race, such as tourists' and locals' aesthetic preference ratings for rural forests [24]. Kaplan and Kaplan [25,26] thought that people inherited an understanding of some features in nature that are aesthetically pleasing. Thus, researchers define forest aesthetics as inherent physical properties and not based on a person's cultural background [27].

While the French philosopher Descartes (1596–1650) argues that reason is the basis of truth and that intuition and deduction are sources of truth, intuition arises solely from the illumination of reason, and deduction is a logical chain of intuitions. Descartes' pervasive influence led to the emergence of the subjectivist view of aesthetics. Descartes paved the way for humans to recognise the role of their subjective feelings in determining aesthetic preferences, rather than viewing aesthetics as an inherent quality of a physical object [13]. In the modern philosophy of aesthetics, this philosophy is known as the subjectivist paradigm, that "visual aesthetic quality is determined according to the observer", and is assessed with psychophysical methods. The British philosophers David Hume (1711–1776), Edmund Burke (1729–1797), and the German philosopher Immanuel Kant (1724–1804) argued that beauty is subjective, describing aesthetics in this paradigm as "the aesthetic in the eye of the beholder", indicating that aesthetics cannot be judged as a whole as it relies on one's personal beliefs and values [6,13,22]. In other words, beauty is cognitive and acquired, and the pleasure of beauty is in the eye of the beholder. This perspective posits that the differences in human responses to aesthetics are caused by cognitive processes altering our perceptions [14]. These studies support the notion that the public perception of aesthetics can be relative and result from socio-cultural constructs, therefore differing depending on

the individual's awareness, experience, or cultural background [24]. This means that, in addition to being physically and sensually stimulating, aesthetic perception is culturally created [28].

These paradigms have historically served as the foundation for how landscape aesthetics has been viewed. Before the last few centuries, the objectivist perspective was the prevailing paradigm. With the development of psychology in the modern era, landscape aesthetics has come to be regarded as being subjective. Nevertheless, numerous studies continue to have difficulty following the notion of aesthetics as subjective or objective approaches. Many efforts have been made to bridge the gap created by separating the subjective and objective approaches to defining aesthetics, and many theorists have advocated the use of objectivity and subjectivity in tandem to develop a comprehensive approach [12,29]. Daniel [12] emphasised that urban forest visual aesthetic values are the result of interactions between biophysical urban-forest physical features (the objectivist paradigm) and human perceptual processes (the subjective paradigm). We recognise that aesthetic definition is founded on both objective and subjective approaches to beauty. Thus, perceiving the aesthetic value of a forest depends on both the physical features of a forest (visual character) and the viewer's perception principles (visual quality). In this study, the previous definition was adopted, along with a reliance on an integrated framework developed by Mundher et al. [2] for the visual aesthetic quality assessment of urban forests using urban forest visual character and urban forest visual quality (Figure 1).

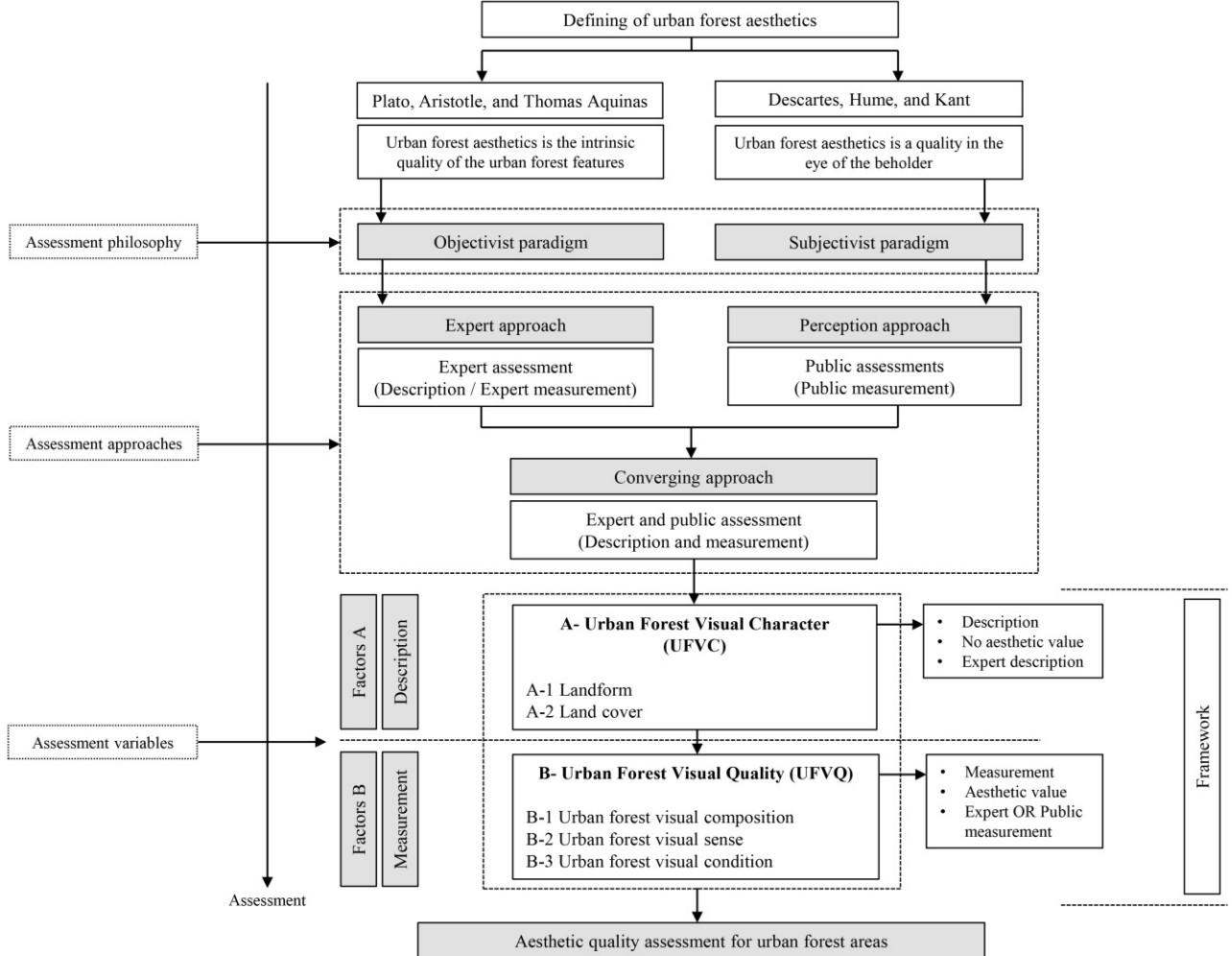

**Figure 1.** A flow of information to get an integrated framework for visual aesthetic quality assessment of urban forest areas. Reprinted from [2].

### 3. Visual Aesthetic Assessment Variables of Urban Forests

Assessing the visual aesthetics of urban forests can be challenging. Nonetheless, Mundher et al. [2] provided a framework based on a process of description and measurement (Figure 2). The categories that influence the descriptive procedure for urban forest visual character are landform and land cover. The measurement process relies on seven variables related to urban forest visual quality (coherence, complexity, legibility, mystery, openness, uniqueness, and cleanliness). Mundher et al. [2] indicated that variables are categorised into three groups: urban forest visual composition, urban forest visual sense, and urban forest visual condition. This study will attempt to define and provide an understanding of the variables that go into describing and measuring visual aesthetic quality of urban forest areas.

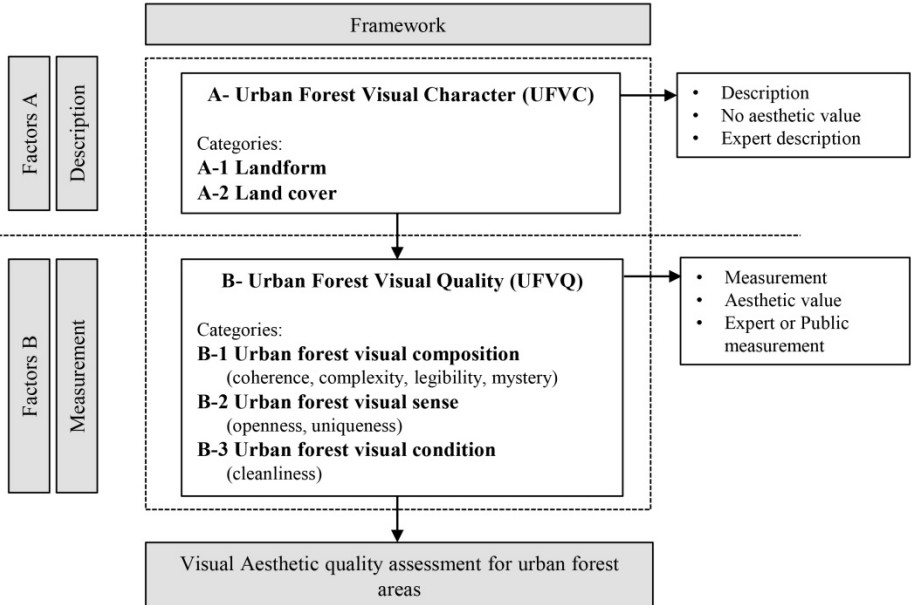

**Figure 2.** Integrated framework for visual aesthetic quality assessment of urban forest. Reprinted from [2].

### 3.1. Urban Forest Visual Character

Urban forest visual character (UFVC) is a particular group of scene elements that create a recognisable scene and give it an identity that separates it from the surrounding area, without necessarily being better or worse [2,30–32]. In other words, the physical components that comprise the scene differ from one another, and gathering them in one unique scene is referred to as "character". Character defines and describes every aspect of the urban forest and gives it a unique sense of place [33]. Therefore, urban forest visual character focussed on forest attributes that offer identity, significance, and a sense of place [33–35]. Proponents of the objective aspect of visual aesthetics regard awareness of urban forest visual character as a crucial key to describing aesthetics [36]. Exploring and knowing the urban forest visual character of any area involves a thorough analysis of the many different variables that contribute to its character. The relationships between these variables create an urban forest's unique character [31]. Identifying forest character is considered to be a descriptive process and definition of the urban forest's visual character. A value-free characterisation method should be conducted with a written clarification of the character's facts for judgment [37,38]. These findings are based on a systematic review that included papers on urban forest visual character [2]; two categories that characterise the urban forest visual character were landform and land cover.

### 3.1.1. Landform

Landform in a landscape indicates the land's topography [31], and is known in science as physiography or geomorphology [39]. Landform variables are easily observable and can be described with a high degree of accuracy. It is relatively stable in forests, and is very effective for land classification, especially for site assessment and forest planning. Topography (landform) is the most distinct aspect defining forests' patterns, scale, and unity [32]. Topography refers to elements on the surface of forests that are related to position and elevation, and are considered the most distinctive forest elements. In addition, topography variation contributes to structural and compositional variability within the forest, leading to various forest types [40]. Consequently, local topographical heterogeneity plays an essential role in forest management [41]. Landform and additional details can help explain and describe the area's topography and identify, as well as the most suitable sites for different land uses. Landform assessment data can be used for urban planning and to support arguments for preserving specific forest locations. Also, landform assessment can help avoid future conflicts of interest and even the loss of valuable locations [42]. Landform has been classified into forest land data for site assessments, including categories such as flatlands, rugged land, undulating land, lowland, mountains, hills, slopes, plains, valleys, and coasts. Therefore, a better understanding of landform character is crucial for urban forest planning, achieving effective, sustainable forest management, and promoting decision-making processes in the urban forest area [43].

### 3.1.2. Land Cover

Land cover indicates physical elements such as green elements, water elements, and human-made elements. The land cover elements are the basic features of the forest that give it its distinctive characteristics and add to its significance as an aesthetic view [33,44]. Forest aesthetic indicators are expressed through land cover element descriptions; for example, forest views featuring water are consistently valued more highly than those without water. Some water features positively affect aesthetics, such as waterfalls, springs, and streams [45,46]. The land cover pattern of an area is considered the product of natural and socio-economic influences, and their use is considered a central component of forest management. Knowledge of land cover is critical to understanding the components of urban forest areas associated with land use and forest transformations, especially urban forests [47]. Describing land cover in urban forest areas involves characterising the forest's content through green elements (perennial forests, native forests, mixed forests, planted forests, and forests with cultural or historical plants). Forests can also be classified based on the prevalent tree species, such as pine or oak, as well as water elements (water forms are often rivers, streams, or lakes), and man-made elements (corridors, entrances, crossings, and structures) [33].

### 3.2. *Urban Forest Visual Quality*

Urban forest visual quality (UFVQ) has recently become an important component of successful planning and forest management techniques [2]. When discussing urban forests as an aesthetic entity, "urban forest visual quality assessment" becomes an essential research topic [10]. Visual aesthetics has historically played a significant part in landscape protection as it is the primary determinant of visual perception [29]. The visual quality of urban forests is one of the essential aspects of the connection between humans and nature, and it is one of the main reasons people visit forest places. It is becoming increasingly important for visual quality to match human aesthetic demands and expectations [48,49]. Visual quality is crucial in presenting individuals with an emotional experience [12]. Furthermore, it is related to human behaviour, the meaning they derive, and the elements that attract their interest. Urban forest visual quality can be described as the "relative aesthetic perfection of a forest" [10,12] assessed from the observer's perspective [13,29]. According to Daniel [12], urban forest visual quality is a common product of the observer's psychological (perceptual, cognitive, and emotional) processes connected with the apparent

forest qualities. However, it is extremely challenging to measure and assess because the dynamic structure of the environment constantly influences user perceptions [49]. Because of these dynamic characters, aesthetic visual quality is likely one of the most challenging phenomena to assess and measure in urban forest areas. In general, visual quality represents the extent to which people's opinions and aesthetic admiration for their surroundings are expressed [10].

For the past 40 years, researchers from various disciplines have been working to understand how the concept of visual quality developed and determine which variables are beneficial. This study is based on a systematic review that included research articles about urban forest aesthetic quality and urban forest visual quality [2]. Seven visual variables were classified into three major groups: landscape visual composition (coherence, complexity, legibility, mystery), landscape visual sense (openness, uniqueness), and landscape visual condition (cleanliness). As a result, this literature study will be used to describe the variables associated with urban forest visual quality.

### 3.2.1. Urban Forest Visual Composition

- Coherence

Coherence is defined as the ability to see and appreciate the pattern inherent in a scene, or how nicely the scene's elements fit together. Coherence represents relatedness, compatibility, and consistency, and is the inverse of chaos [6,50]. The absence of disturbance in the scene can also be defined as visual coherence [8,34]. Balance is a strong manifestation of coherence in the observed environment, especially balancing beauty and understanding [51]. Coherence helps to provide a sense of order and encourages unity. Therefore, fragmentation or chaos inhibits visual coherence [33,52]. There should be an adequate balance or harmony of scenic elements within a scene and a feeling that the individual scenic elements belong together [34]. In Kaplan and Kaplan [26] preference matrix, coherence is one of the four aspects of aesthetic evaluation, and it has an impact on the immediate understanding or sense-making of information emerging from a two-dimensional landscape. In forest aesthetics, coherence describes the unity of a scene, repeated patterns of colour and texture, and the correspondence between land use and natural circumstances [53]. Scenic unity or coherence ensures that a scene will be viewed as suitable and harmonious to its surroundings [54]. Forest coherence has critical implications for future practice, such as the design of aesthetically attractive urban forests with high visual capacity and resilience, forest management, and preservation [11,55].

*Similar terms*: unity, uniformity, balance, harmony, fittingness, compatibility [2].

*Potential indicators*: coherence indicators are correlated with the visual harmony of forest elements present in the forest context as a whole, such as its unity or harmony in colour and texture. This results in a visually pleasing mix of visual quality in forest scenes [27,44,51,54,56].

*Measurement of the variable's value*: coherence value is high if the elements of one whole of the two-dimensional scene are united or harmonious, and vice versa [22,29,52,54,57].

- Complexity

Complexity is the diversity and richness of the elements, features and their interspersions or how intricate the scene is according to the richness of urban forest land cover and landform [11,34]. Overall, complexity refers to the forest's visual aesthetics. According to Kaplan and Kaplan [26], complexity is defined as "the number of various visual elements in a scene; its richness". In a Kaplan matrix, complexity is one of the four aspects of aesthetic evaluation and affects the processing of the immediate exploration or involvement of information emerging from a two-dimensional landscape. In addition, complexity may be divided into ordered and unordered complexity. Ordered complexity provides a scene with visual richness, while unordered complexity can be considered chaotic. Moreover, complexity emphasizes the need for a system and arrangement to demonstrate highly ordered complexity for high aesthetic quality [8,9]. Visual forest complexity or variety

positively affects forest value and public preferences because of the diversity of visual connections between forest elements [50,58]. Complexity has been found to be significant in forest guidelines, forest management, and forest aesthetics preferences [51,55,56,59–61].

*Similar terms*: diversity, variety, richness, heterogeneity [2].

*Potential indicators*: complexity indicators are correlated with scenic forest characteristics such as a variety of objects and types, colour, textures, shapes and masses, shapes and spaces, varied or rugged terrains, or other recognisable attributes that add variety to the visual quality of forest scenes [16,45,54,56,61,62].

*Measurement of the variable's value*: complexity has a high value when the two-dimensional scenes are a variety or diversity of forest features and is not chaotic, and vice versa [29,57].

- Legibility

Legibility refers to how easily a scene may be recognised, understood, and directed [11,29,57]. It is also characterised by how easy it would be to walk around and have visual access without becoming disorientated, and conveys a sense of accessibility [27]. Legibility indicates how easy it would be to navigate and wayfinding the environment to determine where a viewer is at any given time or to return to the starting location [63]. Subsequently, legibility has a positive impact on the safety perception of urban forests. In Kaplan and Kaplan [26] matrix, legibility is one of the four aspects of aesthetic evaluation and affects the processing of the inferred understanding or sense-making of information emerging from a three-dimensional landscape space. Urban forest visual quality scene is determined by the viewers' vision and comprehension, which means that viewers prefer more legible scenes. For example, if the background scenes contain essential features, these features assist people in navigating and comprehending the scene while scattered elements reduce legibility [27]. Legibility has been found to be indicated in urban forest guidelines, urban forest management, urban forests safety, and forest aesthetics preferences [64].

*Similar terms*: clearness, visual access [2].

*Potential indicators*: legibility indicators are correlated with visual access, the number of obstructing elements, and the observer's perception of urban forest elements. Visual legibility and visual coherence are correlated when the scene becomes easy to legible. Likewise, visual legibility contrasts with visual complexity when the scene becomes too complex to be legible. Thus, forests with a high degree of complexity will have low perceived legibility if their constituents are not understood [29].

*Measurement of the variable's value*: legibility has a high value when the three-dimensional scene is clear and interpretable and vice versa [29].

- Mystery

Mystery is the viewer's discovery of elements of the scene as well as the motivation and thrill of finding more hidden information [11]. It is also defined as the extent to which a scene promises more inquiry and curiosity to be discovered and motivates a viewer to walk further [27]. The mystery variable, also known as the "challenge of exploration", is linked to the observer's feeling and experience of exploring a place. In Kaplan and Kaplan [26] matrix, mystery is one of the four aspects of aesthetic evaluation, and it affects the processing of inferred exploration or involvement information emerging from a three-dimensional landscape space. The drive to explore stems from a human need to make sense of their surroundings and the promise of the new information contained therein, although negative feelings could result, for example, if a possible danger is foreseen [25]. Forest exploration may excite people's attention and encourage further exploration but also evoke anxiety and fear if there is too much mystery [57]. Therefore, urban forests should have an element of surprise that is anticipated and produces an artistic sensation without inspiring fear [51]. Mystery has been found to be indicated in urban forest design, urban forest planning management, and forest aesthetic preferences [65].

*Similar terms*: explore the place, inferred exploration [2].

*Potential indicators*: Mystery indicators are correlated with complexity in forest characters with the sensation of being within the scene. Forest characteristics can include spatial topographic heterogeneity and land cover variety [29,57].

*Measurement of the variable's value*: mystery value is high when the three-dimensional scene hides some information or elements from the observer, who can expect the element of surprise without feeling fear or danger, and vice versa [29,57].

### 3.2.2. Urban Forest Visual Sense

- Openness

Openness is the ease with which an observer can reach a wide view of the scene [27,29]. Openness is influenced by the line of sight and viewable area, it is related to the blocking size or breadth of the forest, and it mainly depends on the forest topography [9,34]. In this context, forests are typical natural organisms measured by the openness variable. The presence of openness in the forests improves visibility, which has been linked with the human to forest preferences. Conversely, forest patches shrink when openness rises, resulting in small and isolated areas [8]. The concept of the spatial openness scale is related to evolutionary theories as well as the formal aesthetic, and it incorporates numerous theoretical functions, such as the idea of a "prospect" [66]. According to forest preference research, people prefer open spaces and vistas in an unobstructed way [67]. Therefore, it is a concept that is heavily emphasised in ideas about visual quality and forest aesthetic preference [51,54,61].

*Similar terms*: visibility, enclosure, visual scale, perspective, vastness [2].

*Potential indicators*: openness indicators are correlated with the degree of forest visibility, visual scale (viewable distance and width), and enclosure [8,9,34,61,68].

*Measurement of the variable's value*: openness has a high value when the scene has a wide or panoramic perspective, and the viewer can feel the vastness of the viewshed, and vice versa [29].

- Uniqueness

Uniqueness is the element that gives a forest a high probability of evoking a unique image in a given observer [44,54]. It is defined as the degree to which a scene has been linked to human memory [8]. The forest's uniqueness is described as that quality that distinguishes it and makes it visually striking [9]. The forest's visual sense describes a forest scene with a rare, recognisable quality of uniqueness. Therefore, uniqueness relates to the distinctive qualities of a forest and is better defined as a novelty. It describes forest qualities present in their entirety or by natural and cultural elements that create a clear visual impression in the viewer and render the forest distinctive and memorable [6]. It is believed that familiarity, sympathy, and memory influence uniqueness [69]. Some research has proved this by asking visitors about the beauty of a place. They answered that the tall trees, their roots protruding above the ground, and their location around the waterfalls remained in their imaginations and are unique, whereas if they grew up next to them, they would not [6]. This finding is not surprising, given that the uniqueness variable has been scientifically verified and is an important feature in aesthetic assessments of the forest, particularly in nature tourism [6,28,70].

*Similar terms*: imageability, vividness, sense of place, place identity, distinctive, memorable, attractiveness, familiarity, novelty [2].

*Potential indicators*: Uniqueness indicators are correlated with the forest character and sense of place by reinforcing the image created in the observer's mind. These unique elements can be natural, such as land type and water presence, or cultural symbolic elements [6,8,9,50].

*Measurement of the variable's value*: uniqueness value is high when the scene gives a unique visual impression and is memorable or has unique characteristics such as water or cultural elements and vice versa [8,9,54].

### 3.2.3. Urban Forest Visual Condition

- Cleanliness

Cleanliness is a condition of order and care that contributes to understanding the ideal situation and increases a place's quality through active and careful management [8,27]. The term has frequently been used to convey the strength of management and, especially, the status of physical features [9,34]. Clean or dirty appears to be the most prominent dimension of aesthetic judgments of cleanliness, focussing on preserving a litter-free environment and managing elements that do not appear clean. Cleanliness is primarily concerned with the state of physical elements as well as the degree of human influence. Litter on the ground is the least appealing component of nature-based locations, particularly in urban forests. This is not surprising, given that people do not expect to find litter in natural habitats, and litter in urban forest areas is perceived as "dirty". The public perceives nature-based attractions as more beautiful when there is no evident human intervention [6]. If urban forest features are not founded on and congruent with cultural values, they can be challenging to place and sustain in human-dominated environments. According to this point of view, cleanliness (signs of visible stewardship) is consistent with other aesthetic preference theories that connect sustainability to stewardship [71–73].

*Similar terms*: stewardship, order and care, upkeep, maintenance, safety [2].

*Potential indicators*: Cleanliness indicators have been associated with the terms "condition", "care", and "clean–dirty". It describes whether or not the urban forest has been well-cared for [6,8,9,34].

*Measurement of the variable's value*: Cleanliness value is high if the view exhibits aesthetic care and is clean without exceeding normal limits. If the cleanliness exceeds normal limits, the urban forest will be considered artificial, and vice versa [8,9,34].

## 4. Aesthetic Theories Supporting Visual Aesthetic Assessment Variables of Urban Forests

Visual aesthetics theories are one of the environmental design criteria that influences the protection and sustainable development of forests [2,10]. There are various theories explaining urban forest perception and preferences. These theories are broadly classified as evolutionary theories [25,66] and cultural preference theories [71,74].

The evolutionary theories illustrate how our shared evolutionary past has influenced landscape visual aesthetic perceptions and preferences [66]. According to Davies, the human nature evolutionary theory that our ancestors pursued to increase their chances of survival led to the emergence of aesthetic preferences for particular environments. The assumption is that our ancestors developed preferences for things that were conducive to a preferred lifestyle in their environment. Some physical habitats are more conducive to human flourishing than others, even though humans have repeatedly demonstrated the cognitive capacity and motivation to flourish under much harsher climatic and environmental conditions. These are the preferred habitats because they provide more of what we need and make their attainment less time-, resource-, and energy-intensive. The evolutionary theory of human nature clarified that those who were naturally attracted to these environments and who found them pleasant and appealing would have had a survival advantage over those who were not. This implies that our ancestors' preferred habitats were a result of their environmental adaptability [75]. However, evolutionary theories suggest that all humans share a set of landscape visual aesthetic preferences, both positive and negative. This conforms to the idea that landscape aesthetics is objective and physical, and the beauty of landscapes is an inherent attribute, which means aesthetics is an intrinsic quality. According to this landscape aesthetics approach as a model for urban forests, the genetic basis for preferring urban forest characteristics will remain embedded in humans and explain why people's preferences are similar. In addition, urban forest aesthetics are seen as influencing human development because we respond positively to traits that improve our survival and well-being. The most important theories connected to this idea are the information processing theory [25,26] and prospect–refuge theory [66].

Kaplan and Kaplan [25] presented the information processing theory, which explains the basis of the human need for information and the ability to process it to survive. According to this idea, humans evolved with mental and perceptual capacities for visual processing information that is essential for survival. The theory starts millions of years ago with our ancestors leaving the trees to live in the savannah. There was a need to perceive and comprehend a large amount of visual information, anticipate danger, and process this information quickly [25,26]. As a result, people came to gather information from their surroundings and store it in the form of cognition. This stems from the fact that humans are knowledge-dependent organisms. Therefore, these adaptive biases influence our perceptions and preferences for urban forests. As a result, people should prefer urban forest scenes with features that aid in making sense of information. If the scene contains information that is simple to understand, knowledge can be acquired quickly and thoroughly. Furthermore, the scene must not only convey information, but that information must also be clearly identifiable and assimilable. In contrast, if the scene is confusing and it is challenging to identify information or the observer must perform extensive processing, that information acquisition occurs more slowly and in smaller quantities, making the landscape less preferable. Finally, urban forest preference is composed of two important aspects of human information needs: making sense and involvement. Furthermore, these informational needs are set in a time dimension that addresses both the present or immediate-future or longer-term possibilities [57]. Within this framework, the theory presents an urban forest perception model with four distinct variables: "coherence", "complexity", "legibility", and "mystery" (Table 1). Kaplan's four visual preference variables help understand why people like certain situations and how much they prefer them [25,26].

**Table 1.** The framework of Kaplan information processing theory.

|  | **Understanding/Making Sense** | **Exploration/Involvement** |
|---|---|---|
| **Present or Immediate** Two-dimensional plan | 1-Coherence (How the scene seems to "hang together") | 2-Complexity (The information richness of the scene) |
| **Future or Promised** Three-dimensional space | 3-Legibility (The predicted navigability of the scene upon further exploration) | 4-Mystery (The promise of the scene offering additional information upon further exploration) |

Similarly, Jay Appleton, a British geographer, poet, and researcher, published The Experience of Landscape in 1975. In this book, he investigates questions of "what we like about landscapes and why we like them". In response to these questions, he proposes a theory of prospect and refuge, the roots of which he traces back to Darwinian conceptions of "survival of the fittest". The prospect–refuge theory highlights humans and the landscape's (including the urban forests) dual roles. In addition, the prospect-refuge theory has parallels with the arousal theory, which suggests that an individual experiences an increase in pleasure when viewing an uncertain space or scene [76]. The prospect-refuge theory is based on the behaviour of our earliest ancestors and their relationship to habitats, describing a habitat as a natural place that was required to hunt, gather, or cultivate food, as well as a place where dangerous predators roamed. Under these conditions, it is believed that enclosed spaces promote a sense of safety and relaxation, whereas potential openness is stimulating and exciting. In prospect–refuge theory, the prospect is defined as a place with an unobstructed view, and a refuge is where something can hide. The prospect–refuge theory attempts to explain why certain surroundings seem secure and thus meet basic human psychological needs. Environments that suit basic human psychological needs will provide people with the capacity to observe (prospect) without being seen (refuge). Thus, the prospect–refuge theory explains the preference for being able to "see without being seen". According to this idea, seeing and hiding is essential in evaluating a creature's survival chances. According to Appleton, "the nature of experience is dictated by the essential conditions of life". Appleton contends that human survival instincts are

crucial and that humans derive pleasure from completing all of the activities required for survival. The prospect–refuge idea thus delineates the ability to see without being seen as a manifestation of the survival of the fittest. The prospect–refuge theory states in the urban forest aesthetic that "whether natural or man-made, the strategic value of an urban forest is related to the arrangement of objects which combine to provide these kinds of prospects, and when the strategic value ceases to be essential to survival, it continues to be apprehended aesthetically". As a result, this theory (prospect) supports the "openness" variable, explaining that the urban forests described as open are a source of aesthetic quality [66].

In contrast, cultural preference theories have argued that perception and experience are predominantly dependent on the observer's cultural background and personal attributes. This is compatible with the idea that forest aesthetics are subjective and psychological and forest aesthetics is in the eye of the beholder. These theories confirm that forest aesthetics varies depending on a person's culture and background, and assessments vary from one person to another. The two important theories on this subject that are connected to this idea are the push-pull theory [74] and the aesthetics care theory [71]. The push-pull theory examines whether a destination has distinguishing features that remain in the viewer's imagination. It usually depicts the distinctive view and spirit of the place as motivators for imagination and the place's identity. This finding is unsurprising as scenes with unique elements are more likely to be classified as beautiful and, hence, attractive; this definition serves as the foundation for the push-pull theory. This theory is concerned with the aesthetic judgment of tourists in terms of subjective motivation and their relationship to social action. Some may believe that uniqueness is an objective feature by providing a concept of the ideal scene; however, imagination and familiarity are subjective characteristics that vary from person to person and depend on cultural background. For example, if a person grew up on a small tropical island, they might consider snowy mountains more appealing than the island. As a result, this theory supports the "uniqueness" variable, which explains how urban forests described as motivators for imagination are a source of visual aesthetic quality [74]. On the other hand, Nassauer [71,72] presented the aesthetic care theory, which states that culture is the structure of the urban forest that is readily recognisable by care cues and signifies a continual human presence to care for a forest. Later, the cleanliness or care theory was expanded to include "cues to care" as design features [73]. Thus, this theory supports the "cleanliness" variable, which signifies that a continual human presence to care is a source of visual aesthetic quality.

This manuscript highlights the fundamental contrasts between landscape visual aesthetics theories and interpretations based on biological evolutionary and cultural preference aspects and the distinction between looking at aesthetics as an intrinsic or learned aspect. However, recently we have seen the development of theories of landscape visual aesthetics that rely on a combination of evolutionary and cultural theories to arrive at an integrative conceptual framework that can be incorporated as a guideline in the process of assessing the visual aesthetic qualities of urban forests. This combination explains landscape visual aesthetics are conceived and preferred by assuming that all humans have the same genetic basis that has been altered by cultural influences and personal experiences. Finally, we agree with and support the integrative theoretical framework, and we believe that due to our shared evolutionary history, humans have a standard set of urban forest visual aesthetic features with preferences that change according to cultural and personal variances (Figure 3).

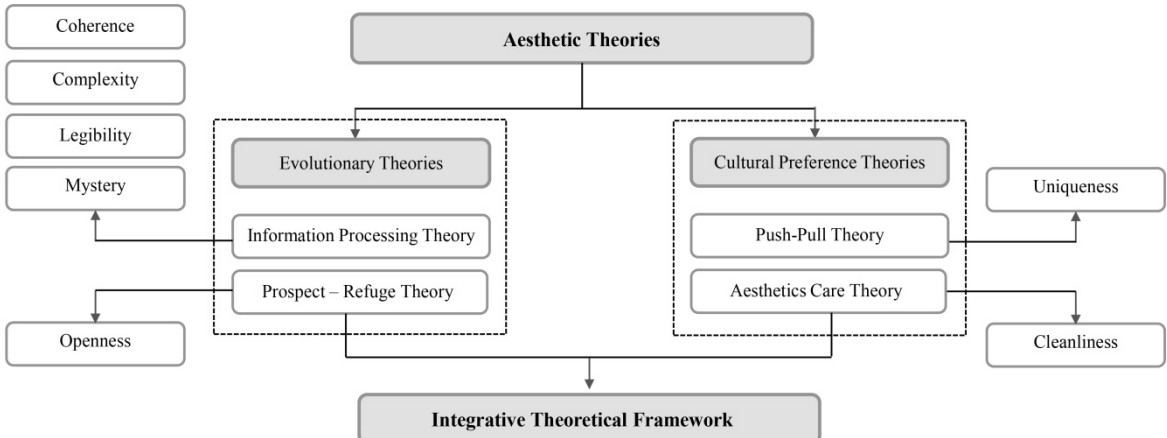

**Figure 3.** Aesthetic theories framework provide support to visual aesthetic quality variables of urban forests.

## 5. Discussion

The visual aesthetic quality impact of urban forests is crucial because of their impact on our daily lives, particularly in terms of health, tourism, and economics. This qualitative research aims to establish a foundation for understanding the visual aesthetic of urban forests by understanding the variables, which will assist urban forest managers and decision-makers in managing and protecting the visual aesthetic of urban forests. Also, visual aesthetic variables will assist urban forest researchers in future quantitative research on visual aesthetic assessment. In addition, defining and understanding the variables that go into determining the visual aesthetic quality of urban forests will enable decision-makers to make accurate decisions. Subsequently, understanding the variables is essential for preserving and sustainably managing aesthetically pleasing urban forests. Therefore, visual aesthetic variables should not be overlooked while developing long-term sustainability plans for urban forest areas. To grasp the variables essential in describing and evaluating the aesthetics of urban forests, one must first understand aesthetic philosophy. After reviewing all of the selected documents, we found that all of them adhere to philosophical approaches, some of which may be traced back to the objective approach, which defines and explains urban forest aesthetics through the classification of urban forest visual characters. These characteristics serve as the foundation for defining and describing urban forest visual aesthetics. Another set of visual aesthetic assessment researchers prioritise the subjective approach and aim to evaluate the aesthetic using the urban forest visual quality variables. Their work is predicated on the concept that visual quality is a component of aesthetic quality. Many opinions and variables on which scholars differ exist concerning the variables of visual aesthetic quality. However, we relied on previous research in the classification of variables [2], which consists of seven main variables divided into three categories (coherence, complexity, legibility, mystery, openness, uniqueness, and cleanliness). We recognise that the two approaches differ philosophically, but we endorse a framework for a comprehensive approach to describing and measuring aesthetics that incorporates objective and subjective factors through urban forest visual character and urban forest visual quality.

The process of describing aesthetics requires an expert who can describe and define aesthetics through urban forest visual characters. However, the process of aesthetics measurement requires the incorporation of local people's preferences and judgments as a foundation for aesthetics measurement experts. As a result, we must discuss these variables and their theoretical foundations to sufficiently grasp the value of each variable. This study is based on four of seven main variables from Kaplan and Kaplan [26] for assessing aesthetics from the standpoint of information processing theory and on two-dimensional scenes that present coherence (understanding), and complexity (exploration) and three-

dimensional scenes that present legibility (understanding) and mystery (exploration). Thus, coherence and legibility are two linked variables that reflect unified and straightforward information understanding; the difference between them is in the dimensions, which leads us to conclude that coherence in three-dimensional scenes leads to legibility. According to trials and research that depend on the viewer's preference, many researchers have proved that coherence is a crucial and influential variable in determining the visual aesthetics of forests [6,22,29,49]. At the same time, others indicated that legibility is less preferred [57,65], possibly due to excessive unity in the scenes, which gives a feeling of depression. However, if the background scenes contain significant landmarks, these landmarks increase both legibility and preference.

Meanwhile, complexity and mystery are two linked variables that indicate information exploration, and the difference in dimensions leads us to conclude that complexity in three-dimensional situations leads to ambiguity. That is, complexity can produce ambiguous experiences in two-dimensional scenes [51]. Many researchers have identified complexity as one of the significant criteria for aesthetic judgment [6,65]. The presence of a diverse blend of interconnected elements generates a high aesthetic preference. Although an increase in exaggerated diversity decreases preference [60], this could be because of the viewer's inability to focus on many of the elements, and so the complexity is associated with orderly diversity that is not chaotic. Others noted that, while mystery was related to the human will to explore, it was also associated with fear and danger, which proved to be less preferred [29,57,65]. This can occur due to the concealment of certain aspects or the existence of certain impediments to scenes, which can make the area feel more threatening. Thus, in forest scenes, we expect that fear and danger will increase as the degree of mystery increases, and preference will decrease. However, Subiza-Pérez et al. [51] argue that mystery is given the highest level of preference in urban forests, demonstrating that mystery is a significant and accepted aesthetic component of the urban forest. As a result, researchers have not paid enough attention to understanding the degree of mystery and its impact on preference in urban forests, raising the question of whether the degree of mystery elicits positive or negative feelings.

Despite everything indicated in these studies, which should investigate urban forest areas to give specific results, Kaplan and Kaplan [26] were creative in constructing a matrix of four variables that serves as the foundation for gauging aesthetics. Kaplan and Kaplan [26] variables are known as evolutionary theory variables. The openness variable was found to be yet another variable that adheres to evolutionary theories. Rosley et al. [27] proved that these variables are less sensitive to cultural and personal background than others by relying on similarities between experts' and the general public's preferences, which can be attributed to the intrinsic values and information that humans need to survive. The study also discovered that forests with a high level of openness and a high sense of order inspire more positive feelings and higher preferences [77]. Liu and Schroth [52] revealed a strong correlation between openness and the Kaplan variables, explaining that when a space is open and clear, people see it as having greater coherence and legibility and less complexity and mystery.

On the other hand, other variables indicated that a person's background considerably influences their preference. As a result, determining visual quality depends on the observer's personal and cultural qualities. One of these variables is uniqueness, which has been proven to influence aesthetic preferences [49,54]. Notably, there is a strong association between singularity and preference in the aesthetic quality of waterscapes. The water element is one of the distinct elements that influence people's preferences, and it is often the dominant element [78]. Furthermore, if urban forests feature distinct cultural or symbolic components, these scenes will be preferred. Features connected to uniqueness are more essential in urban forest aesthetics for tourists than other variables related to coherence, complexity, and others [6]. According to this theory, if unique features are available to rapidly and effectively regulate the aesthetics of urban forest areas, elements such as novelty, vividness, sense of place, and distinctiveness are more preferred. Cleanliness is the

last variable assigned by the theory of aesthetic care [72]. Landscape perception and choice are thought to be influenced by cleanliness [34]. People generally embrace the concept of cleanliness because they do not expect to see obvious filth in forested areas. On the other hand, cleaning procedures involving removing of fallen branches and leaves are occasionally disliked for safety and aesthetic reasons. Nevertheless, some branches and leaves from urban forests may be left for reasons related to the urban ecosystem [79]. Greatly increasing cleanliness will result in the scenes becoming artificial landscapes, which people do not enjoy. Eventually, the public considers nature-based attractions more aesthetic when there is no considerable and evident human interference in urban forests.

## 6. Conclusions

The visual aesthetic quality of urban forests based on aesthetic theories gives an integrated framework and suitable explanation for each variable, which helps to assess the visual aesthetic quality of urban forest areas. Defining and understanding variables for visual aesthetic quality can be a valuable first step in the assessment of the visual aesthetics of urban forests. In practice, the framework can give a more straightforward and sustainable idea for local governments to make the proper judgments to protect the aesthetic quality of urban forests. However, we still need to explore the applicability of these variables and how to assess them for urban forests. Furthermore, we think that the weights and importance of these variables are unequal, so we encourage researchers to determine them. However, researchers cannot begin to do so without conducting a preliminary study to define and understand the variables for the visual aesthetic quality assessment of urban forests. As result, this research provides a foundation of visual aesthetic variables of urban forests. We believe that this study's findings have identified key variables for visual aesthetic assessment in urban forests that will assist researchers, forest managers, communities, and non-governmental organisations in protecting and managing the visual aesthetic of urban forests.

**Author Contributions:** Conceptualisation, R.M. and S.A.B.; data review, R.M., M.A.-H. and H.G.; writing—original draft preparation, R.M. and A.A.-S.; review and editing, S.A.B., M.J.M.Y. and S.M.; visualisation, R.M.; supervision, M.J.M.Y., S.M. and A.A. All authors have read and agreed to the published version of the manuscript.

**Funding:** This research received no external funding.

**Institutional Review Board Statement:** Not applicable.

**Informed Consent Statement:** Not applicable.

**Data Availability Statement:** Not applicable.

**Conflicts of Interest:** The authors declare no conflict of interest.

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
