# Peer review of "Visual Aesthetic Quality Assessment of Urban Forests: A Conceptual Framework"

_urbansci, doi:10.3390/urbansci6040079_

Round 1
Reviewer 1 Report
Dear authors,
This research is quite interesting and well-developed. It fully explained visual aesthetic factors and variables with a thorough explanation of visual aesthetic theories. Below are a few suggestions for a revision:
1. For the introduction, in the beginning, the meaning of studying forest visual aesthetics can explain more (lines 41-43, the impacts on well-being, health, and economy.)
2. The section titles are too long, please make them be shorter and more specific.
3. For the in-text citation method, should mention the surname of the authors, not “Ref. [7] thought that…”
4. For the sequence, is it better to first discuss “Define Aesthetic Theories Provide Support to Visual Aesthetic Quality Variables of 389 Urban Forests” and then “Defining and Understanding the Variables for Visual Aesthetic Quality Assessment 121 of Urban Forests”?
5. Some spacing problems in the manuscript should be resolved.
6. For the discussion part, can you discuss more how the visual aesthetic variables can assist urban forest researchers, urban forest managers, and decision-makers?
Thank you
Author Response
All answers are accessible in the PDF file attached.

Reviewer 2 Report
It is a kind of review paper basing strongly on literature and aesthetical theories analysis. The Authors undertook the difficult issue of aesthetic evalution of the landscape, in this case study - visual evaluation of urban forests. As a result of this anlysis, the Authors proposed, in fact, an oryginal methodic approach, how to do such aesthetic evaluation. The problem of visual/aesthetic evaluation of the landscape is not often presented and discussed, as it is not a simple thing, depending on many factors and causing methodological troubles (especialy the occurrence/ use of objective and subjective factors/criteria). So, I found the article very interesting and necessary.
In the presented methodical approach, the Authors described criteria of aesthetic evaluation, skillfully combining objective and adjective factors. The paper is mostly a scientific discussion on evaluation criteria and methodic approach to solve the problem of aesthetic assessment of the landscape.
The Authors have presented (proposed) criteria of aesthetic evaluation of urban forests. I am not sure, whether the naturalness of forest ecosystems is included in these criteria - ? I also feel the lack of explanation how to use these criteria - how to evaluate under each ctiterion (from low to high evaluation - what gradation to use in the evaluation). It will be more valuable, if the Authors complete better their methodic proposition and arguments.
Detailed comments:
In my opinion, the Authors should not double the adjectives: "visual aesthetic" (evaluation) in the title and in the whole article. It is "butter butter". In my opinion, use one adjective: aesthetic evaluation or visual evaluation (may be used alternatively).
In the Key Words: rather: Aesthetics (not: visual aesthetic).
The structure (subsections) of the paper is not typical, but interesting and justified. Perhaps it will be better to combine subsections 2 and 3 into one: "Methods", will not it?
In general: The article is worth publishing after minor corrections. It is very interesting and partly original, as regards methodic considerastions and proposition of methodic approach (mainly in the range of selection of criteria of aesthetic evaluation).
Author Response

(The authors gave the same response as above.)

Reviewer 3 Report
This article addresses an important question, and that is about the visual aesethetics of urban forests. Since urban forests now seem to have become an important attribute in cities, I had hoped this article might enlighten about what makes an urban forest aesthetically pleasing.
I question using totalizing human nature evolutionary theory variables and preference for open forests, for example. the citations do not include any indigenous peoples with regard to their preferences. the example of living on a small tropical island and the preferring snowy peaks seems entirely fictional. I grew up in the Western United States, and do not care for urban forests, I find them smothering. these types of statements out of no where (the small island) are not substantive contributions, rather inventions.
I suggest the authors need to define what they mean by urban forest. the article uses the term natural forest, but few cites have natural forests within them, so natural forests are the topic, then the authors are dealing with a few number of cities, and probably cities largely in the tropics?
if not, then the question of humanly designed urban forests is not addressed and should be -- who is deciding forest composition for example, and the spatial arrangements within the forest? the concept of cleanliness may not be conducive to a healthy natural forest. Are twigs and leaves 'filth'? German urban forests are not manicured, but twigs and leaves are not considered 'filth'.
I am also a bit surprised that the only referents for forest aesthetics seem to come from the West. Certainly western views of aesthetics cannot be the only ones? Plato, Aristotle, Aquinas etc . . are not universal philosophers. the article is missing Confucuis, Lao Tzu and many others, including some western philosophers such a Augustin Berque who writes about Japan.
The authors hope that this framework can help local governments to make proper judgements about the aesthetic quality of forests. I would suggest that the authors might take a step back and examine the question of what they mean by urban forest; the conditions under which they might arise or not; the enormous climatic and geographical variation of cities (
Reykjavík vs. Los Angeles) and think more deeply about what an urban forest means in those cases. Further, some cities are dense, some are not, some cities are wealthy, some are not.
The topic is of interest, the approach so totalizing as to be not implementable.
Author Response

(The authors gave the same response as above.)
